# Analytical Study on Reinforced Concrete Columns and Composite Columns under Lateral Impact

Xianhui Li [1,2], Yao Yin [1], Tieying Li [1], Xiang Zhu [3,*] and Rui Wang [1]

1. College of Civil Engineering, Taiyuan University of Technology, Taiyuan 030024, China
2. College of Architectural Engineering, Shanxi Vocational University of Engineering Science and Technology, Jinzhong 030600, China
3. School of Electric Power, Civil Engineering and Architecture, Shanxi University, Taiyuan 030013, China
* Correspondence: zhuxiang@sxu.edu.cn

**Abstract:** This study investigates the lateral impact responses of reinforced concrete (RC) and composite columns through dynamic nonlinear analysis using LS-DYNA. The simulation results were first validated against experimental results performed earlier on four different cross sections. The finite element analysis results show that the simulation results of LS-DYNA can predict the experimental results well and can be used for further parametric analysis. The overall impact resistance of the four new composite columns is significantly better than that of RC columns. Among the composite columns, the solid concrete-filled double steel tube (S-DS) column has the best impact resistance with higher impact plateau force and smaller mid-span deflection under the same test conditions. It was found that the impact response process of all types of composite columns was similar. Finally, parametric analysis of the composite columns is performed to study the influence of load, material and other related parameters on the impact response of the composite columns. The results provide new information on the impact response of composite columns and the influence of materials and load parameters. The study provides a basis for the design and analysis of composite columns under lateral impact loading.

**Keywords:** RC columns; composite columns; lateral impact; numerical simulation; whole-process analysis; parameter sensitivity analysis





## 1. Introduction

The collision load effect on structural components has to be considered in structural design and sometimes considered in whole life cycle analysis. Relevant reports on collision accidents are increasing year by year [1–5]. It is well-known that the compressive strength of concrete is much higher than its tensile strength. Furthermore, the compressive strength is enhanced under bi-axial or tri-axial restraint. For structural steel, the tensile strength is high while the shape may buckle locally in compression. In concrete-filled steel tubular members, steel and concrete are used such that their natural and most prominent characteristics are taken advantage of. The confinement of concrete is provided by the steel tube, and the local buckling of the steel tube is improved due to the support of the concrete core. To ensure the safety of buildings and other structures under impact loading, concrete-filled steel tube (CFST) composite columns were used to improve the impact resistance of new structural columns. In the past few decades, CFST has been extended to other forms of composite columns [6–8], such as R-ST columns, S-ST columns, H-DS columns and S-DS columns. These composite columns are expected to have improved impact resistance to reduce the losses caused by accidental collisions.

The mechanical properties of composite columns under impact have been studied by a number of researchers. Aghdamy et al. [9,10] studied the impact resistance of hollow concrete-filled double steel tube columns using parameter sensitivity analysis. Results show that the deflection is significantly affected by axial load. The impact velocity, slenderness

ratio and thickness-to-diameter ratio of steel tube are the key factors in determining the dynamic response of CFST members as these are the parameters that determine the impact energy of the loads and stiffness of the members. Zhu et al. [11,12] studied the impact resistance of steel-reinforced concrete (SRC) columns and steel-reinforced concrete-filled circular steel tube (SRCFST) columns. The results show that the impact resistance of SRCFST columns is better than that of SRC columns under the same mass. Wang et al. [13] studied the behaviors of CFST under lateral impact by experimentation and finite element methods. The results show that the critical failure energy increases with an increase in the constraint factor, and the axial force has a significant influence on the lateral displacement of specimens. Xu et al. [14] and Wang et al. [15] studied the impact behaviors of carbon fiber-reinforced polymer (CFRP)-reinforced RC columns by experimental and numerical methods. The results show that CFRP-reinforced RC columns can effectively change the lateral impact failure mechanism from shear to bending, which can effectively reduce impact damage. Han et al. [16–18] studied the impact resistance of a CFST column and a CFST composite column by experimental and finite element methods. The results showed that the CFST column had good impact resistance, while the concrete in the outer layer of the composite column was seriously damaged; inner steel tube only had limited flexural deformation. Wang et al. [19,20] carried out lateral impact tests on 31 hollow concrete-filled double steel tube columns using a drop hammer device. The results showed that CFDST members had good ductility under impact loading, and the sandwich steel tube had an obvious restraint effect on concrete. Goldston et al. [21] investigated the behavior of glass fiber-reinforced polymer (GFRP) bar-reinforced high-strength concrete and ultra-high-strength concrete under impact loading. Swesi et al. [22] studied CFRP strengthening on the response of RC columns under lateral impact loads.

Recently, a total of 30 columns (6 RC columns and 24 composite columns) were tested under low-velocity impact by Zhu et al. [23]. The test results show that the change in the axial compression ratio has an important effect on the impact resistance of specimens. In this paper, LS-DYNA software is used to describe the nonlinear finite element analysis (FEA) of RC and composite columns under transverse impact. The FEA model is validated with experimental results. Then, the whole process of impact damage on the composite column is analyzed. The validated model is then used to conduct an extensive parametric study to investigate the effects of impact height, impact mass, unconfined compressive strength of concrete, yield strength of outer and inner steel tubes, diameter–thickness ratio of outer and inner steel tubes, influence of H-shaped steel on impact force and mid-span residual deflection.

## 2. Finite Element Model

### 2.1. Composite Column Parameters

The validation of the models is based on an impact test of composite columns [23]. This numerical study simulates the impact resistance of composite columns with five different cross sections (shown in Figure 1A) under axial compression ratios of 0, 0.3 and 0.5, resulting in 15 combinations, which are calculated as $N_0/N_u$, where $N_0$ is the axial force applied in the test and $N_u$ is the axial ultimate capacity of the specimen obtained from references [24–27]. The parameters of the columns are shown in Table 1 and the dimensions of the different sections are shown in Figure 1A [23]. To compare the impact resistance of members with different section types, the composite columns and reinforced concrete columns are designed to have the same outer diameters and use the same amount of steel. The outer diameter of the steel tube is 114 mm and the wall thickness is 2 mm. For solid concrete-filled double steel tube (S-DS) and hollow concrete-filled double steel tube (H-DS) sections, the inner circular steel tube diameter is 50 mm and the wall thickness is 2.5 mm. The size of the cross-sectional structural steel of S-ST columns (Figure 1(Af)) is 57 mm × 14 mm × 2 mm × 3 mm ($h \times b \times t_1 \times t_2$). The longitudinal reinforcement diameter ($D$) of steel tubular-encased reinforced concrete (R-ST) and reinforced concrete (RC) members is 6 mm, and the stirrup diameter ($d$) is 4 mm. There is a 300 mm long plastic hinge zone at both ends of the specimen, where the

stirrup spacing is 100 mm. Stirrup spacing is 200 mm outside the plastic hinge zone. The member length is 1800 mm for all.

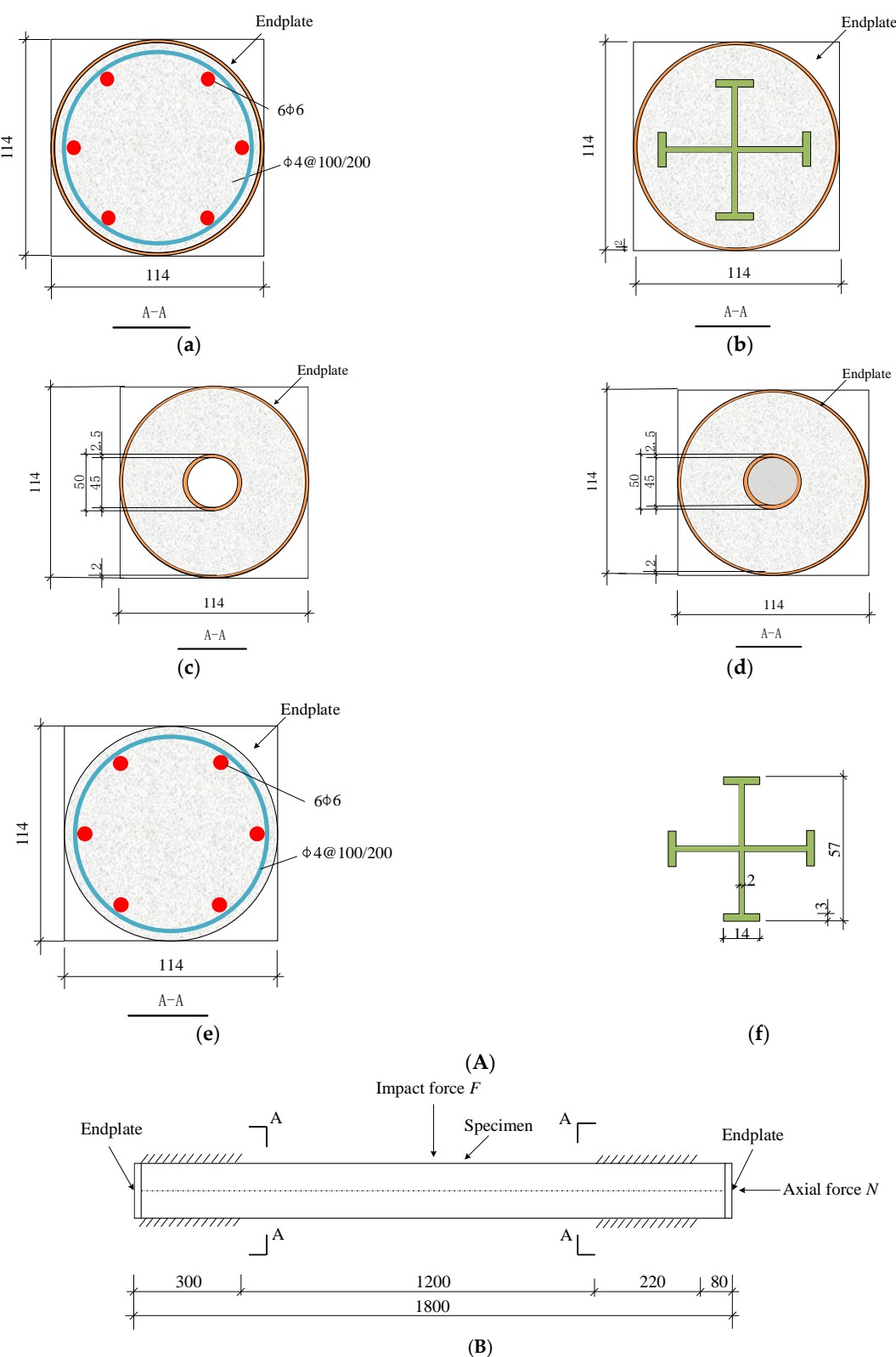

**Figure 1.** (**A**) Dimensions of the specimens (unit: mm). (**a**) Section of R-ST columns; (**b**) Section of S-ST columns; (**c**) Section of H-DS columns; (**d**) Section of S-DS columns; (**e**) Section of RC columns; (**f**) Section dimensions of the double H-shaped steel. (**B**) Plan view of the specimens.

**Table 1.** Specimen information table.

| The Name of the Specimen | ID | $N_0$ (kN) | $v_0$ (m/s) | $m_0$ (kg) | $F_{stab}$ (kN) Test | $F_{stab}$ (kN) FEA | $t_d$ (ms) Test | $t_d$ (ms) FEA | $\Delta_{res}$ (mm) Test | $\Delta_{res}$ (mm) FEA |
|---|---|---|---|---|---|---|---|---|---|---|
| Reinforced concrete (RC) | RC1 | 0 | 4.43 | 206.65 | 27.40 | 30.56 | 46.10 | 44.72 | 50.90 | 37.70 |
| | RC2 | 59 | 4.43 | 206.65 | 22.45 | 22.09 | 45.45 | 31.52 | 80.88 | 63.93 |
| | RC3 | 99 | 4.43 | 206.65 | 23.30 | 11.78 | 61.15 | 46.04 | — | — |
| Steel tubular-encased reinforced concrete (R-ST) | R-ST1 | 0 | 9.39 | 206.65 | 135.24 | 129.33 | 23.60 | 21.19 | 56.72 | 60.82 |
| | R-ST2 | 150 | 9.39 | 206.65 | 126.13 | 106.30 | 24.40 | 23.91 | 65.06 | 66.99 |
| | R-ST3 | 250 | 9.39 | 206.65 | 104.70 | 97.66 | 27.55 | 24.33 | 73.87 | 77.53 |
| Steel-reinforced concrete-filled steel tube (S-ST) | S-ST1 | 0 | 9.39 | 206.65 | 131.45 | 109.26 | 23.35 | 27.45 | 63.65 | 59.12 |
| | S-ST2 | 190 | 9.39 | 206.65 | 118.69 | 88.68 | 24.90 | 29.90 | 75.51 | 69.22 |
| | S-ST3 | 315 | 9.39 | 206.65 | 88.97 | 75.26 | 31.15 | 33.75 | 89.86 | 83.57 |
| Hollow concrete-filled double steel tube (H-DS) | H-DS1 | 0 | 9.39 | 206.65 | 144.22 | 122.80 | 22.30 | 23.20 | 52.14 | 53.55 |
| | H-DS2 | 170 | 9.39 | 206.65 | 129.50 | 108.11 | 23.40 | 24.25 | 70.05 | 61.67 |
| | H-DS3 | 285 | 9.39 | 206.65 | 112.96 | 105.96 | 25.55 | 24.05 | 80.17 | 75.67 |
| Solid concrete-filled double steel tube (S-DS) | S-DS1 | 0 | 9.39 | 206.65 | 125.34 | 112.95 | 21.50 | 24.25 | 57.82 | 51.14 |
| | S-DS2 | 215 | 9.39 | 206.65 | 133.90 | 110.50 | 23.20 | 23.45 | 68.17 | 58.85 |
| | S-DS3 | 355 | 9.39 | 206.65 | 113.27 | 112.80 | 24.65 | 22.85 | 76.37 | 69.54 |

The impact mass of the impact test is 206.65 kg, and the impact height of the hammer is 4.5 m, corresponding to an impact speed of 9.391 m/s. When the impact velocity is 9.391 m/s, the damage to the RC component is serious and has no analytical value, so the impact velocity of the RC component is set to 4.43 m/s. The test equipment is illustrated in Figure 2. During the test, the column is allowed to deform freely along the axial direction. The axial load was maintained at designed axial load ratios of 0, 0.3 and 0.5. The boundary conditions of the composite column in the test are fixed at both ends for rotations, and the effective length ($L_0$) of each specimen is 1200 mm (shown in Figure 1B).

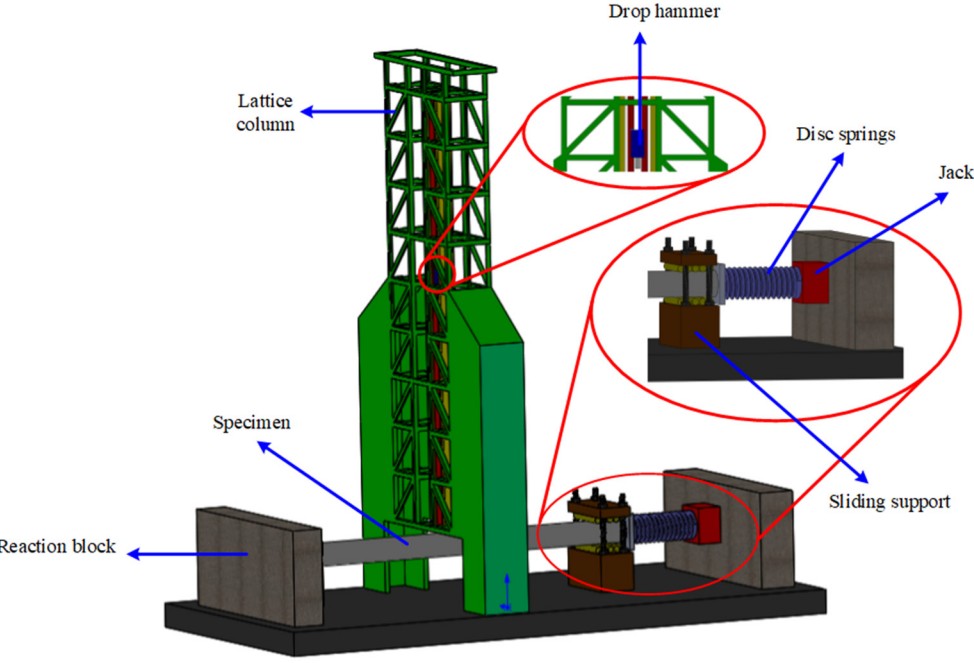

**Figure 2.** Test setup.

### 2.2. Finite Element Mesh and Contact Definition

In the finite element model, the outer and inner steel tubes and H-shaped beams are simulated with four-node SHELL163 elements, and the thickness of the steel tube has five integral points to capture the local bending of the steel tube. The filled concrete and drop hammer were modelled using eight-node SOLID164 elements. Through the study of mesh convergence, the appropriate density and mesh element are determined to ensure the simulation has sufficient accuracy and efficiency, and ensure that the hourglass energy is less than 5% of the total impact energy [28]. Figure 3 shows the typical finite element model with a minimum mesh size of 5 mm.

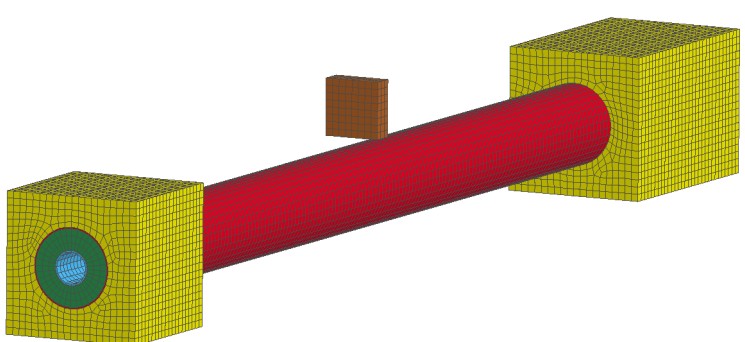

**Figure 3.** Schematic diagram of finite element model.

In the tests, it was observed that there is no slip between concrete and steel tubes, steel bars and H-shaped beams. Thus, the slip between steel tubes, steel bars and concrete is not considered in the simulation and the fluid–structure interaction (* CONSTRAINED LAGRANGE-IN-SOLID) method is adopted to bond the steel bars and concrete. All other contacts adopt automatic face contact (* CONTACT_AUTOMATIC _SURFACE_TO_SURFACE). The friction factor of the Coulomb friction model was taken as 0.15.

### 2.3. Materials Properties

Concrete is modelled by continuous cap material (* MAT-CSCM-CONCRETE), which is a cap model with a smooth intersection between the shear yield surface and the hardening cap. The initial damage surface coincides with the yield surface. Rate effects are modelled with viscoplasticity. The material model is widely used in numerical simulation due to its simplified input parameters and good simulation results. The unconfined compressive strength of a cylinder used in the simulation is 32.94 MPa, which is calculated as the average compressive strength of a concrete cube measured on the test day multiplied by 0.79 [29]; this coefficient is caused by the different calculation methods of the compressive strength of concrete in China [24] and the United States [30]. The size of the concrete aggregate is set to 5 mm.

Steel bars and steel plates are modelled by kinematic hardening plasticity with the option of including rate effects (* MAT_PLASTIC_KINEMATIC). This model is suited to model isotropic and kinematic hardening plasticity. The strain rate is accounted for using the Cowper and Symonds model [31] which scales the yield stress with a factor. It is a very cost-effective model and is available for beam (Hughes–Liu and Truss), shell and solid elements. The Young's modulus ($E_s$), yield strength ($F_y$), ultimate tensile strength of steel ($F_u$) and final failure strain ($\delta$) of the material model parameters were obtained from the experimental test listed in Table 2, and the values of strain rate parameters D and p were set as 40.4 and 5, respectively [30].

**Table 2.** Material information sheet.

| Steel Type | Diameter of Steel Bar or Thickness of Steel Plate (mm) | $F_y$ (MPa) | $F_u$ (MPa) | $E_s$ (MPa) | $\delta$ |
|---|---|---|---|---|---|
| Stirrup | 4 | 603.5 | 770.5 | 2.23 | 0.104 |
| Longitudinal bar | 6 | 529.2 | 681.1 | 2.17 | 0.206 |
| Outer steel tube | 2 | 330.1 | 396.7 | 2.21 | 0.192 |
| Inner steel tube | 2.5 | 330.4 | 398.4 | 1.75 | 0.235 |
| Web of H-shaped steel | 2 | 295.1 | 356.2 | 1.95 | 0.317 |
| Flange of H-shaped steel | 3 | 295.3 | 362.4 | 1.92 | 0.322 |

The drop hammer is modelled by rigid-body material (* MAT_RIGID). Mechanical properties of materials and degrees of freedom (excluding degrees of freedom in the direction of impact) are constrained. The impact velocity, impact mass and contact size of the drop hammer and specimen are consistent with the test.

## 3. Validation of the FEA Model

Figure 4 shows a comparison of predicted and observed failure modes for RC columns and composite columns. The residual deformations of the model include the overall flexural deformation, local buckling deformation of the outer steel tube and fractures. A more detailed comparison in terms of forces and displacements is presented in this section.

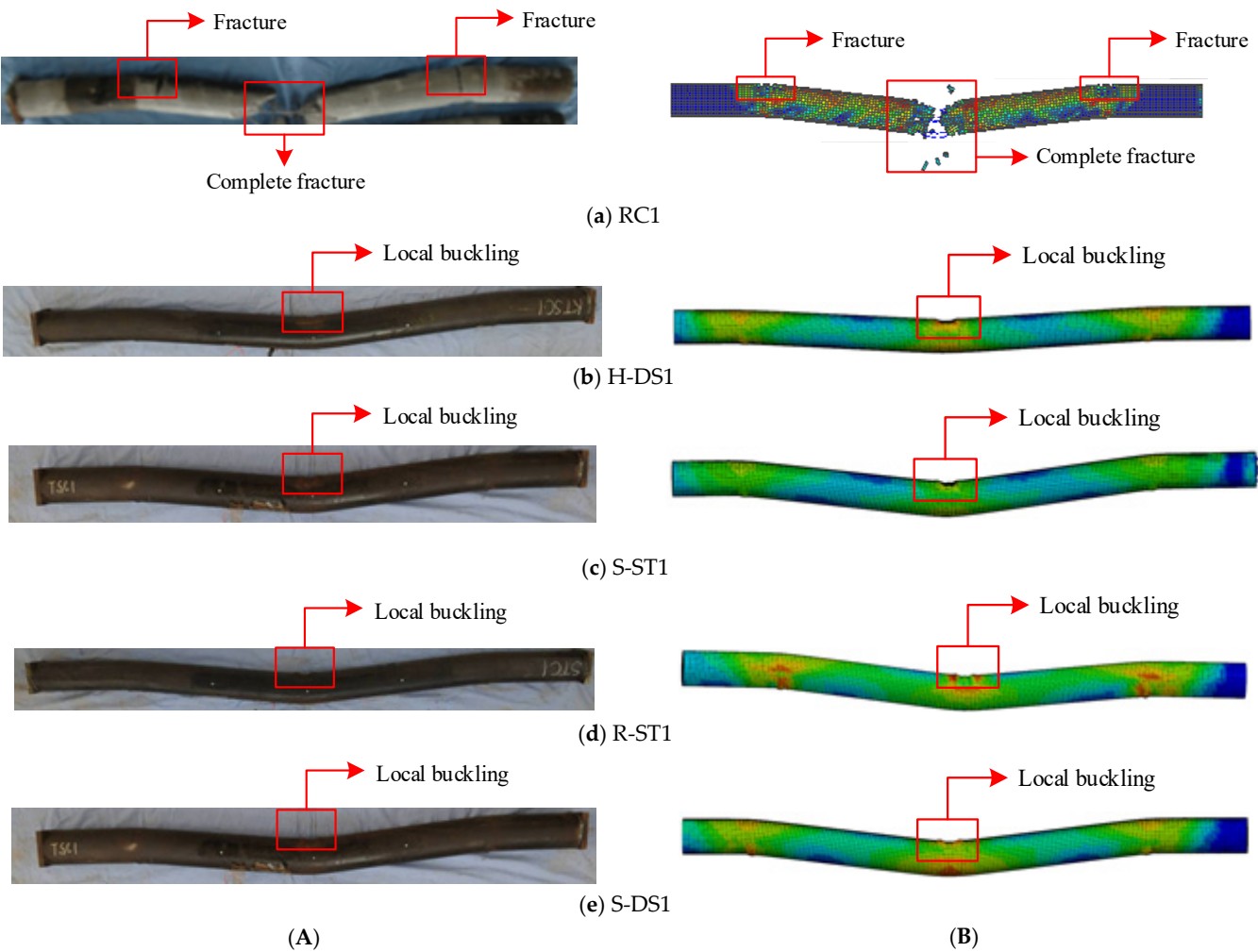

**Figure 4.** Comparison of observation and prediction of typical failure modes of RC columns and composite columns. (**A**) Observed in the test and (**B**) Predicted in FEA.

Figure 5 shows the comparison between the predicted and measured impact force–time histories of five composite column specimens. It can be seen that, except for the RC column, all composite sections fit well with the test results. This is because the RC column suffered severe impact damage and the CSCM concrete material model of LS-DYNA cannot accurately simulate concrete damage.

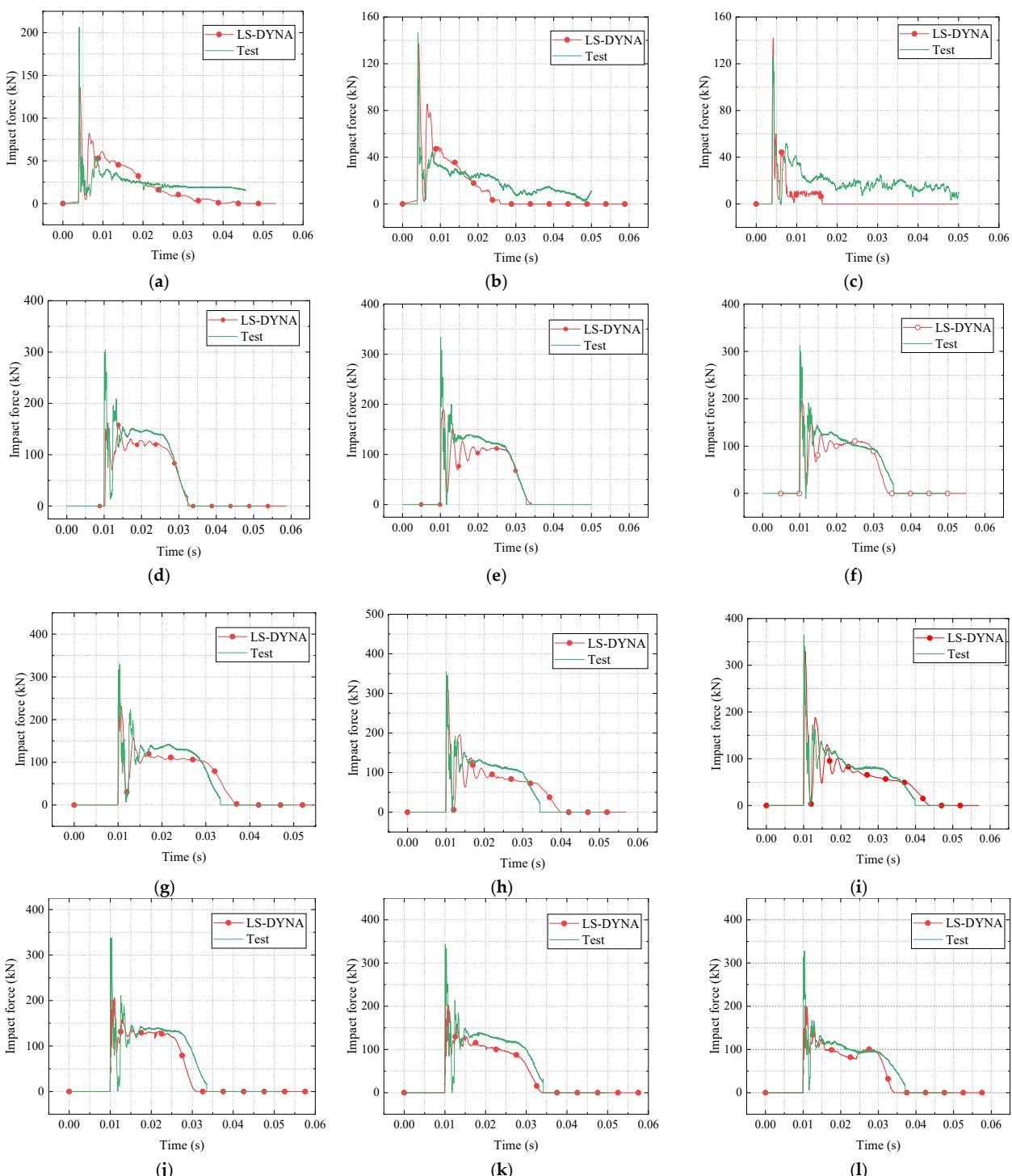

**Figure 5.** *Cont.*

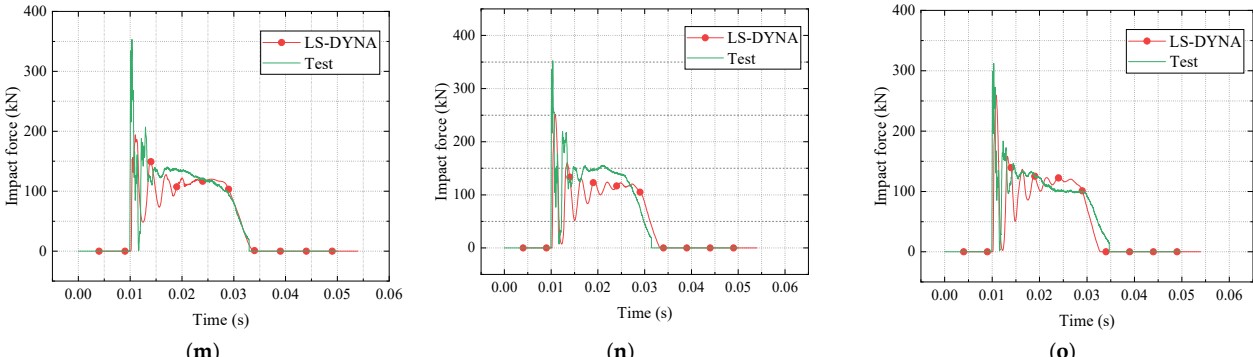

**Figure 5.** Impact force–time history curve. (**a**) RC; (**b**) RC0.3; (**c**) RC0.5; (**d**) H-DS; (**e**) H-DS0.3; (**f**) H-DS0.5; (**g**) S-ST; (**h**) S-ST0.3; (**i**) S-ST0.5; (**j**) R-ST; (**k**) R-ST0.3; (**l**) R-ST0.5; (**m**) S-DS; (**n**) S-DS0.3; (**o**) S-DS0.5.

Figure 6 shows the comparison of predicted and measured mid-span displacement–time histories of five column specimens with different cross sections. It can be seen that, consistent with the impact force–time history curves, the mid-span displacement of the simulated RC column deviates from the test result due to the limited capacity in simulating severe concrete damage.

**Figure 6.** *Cont.*

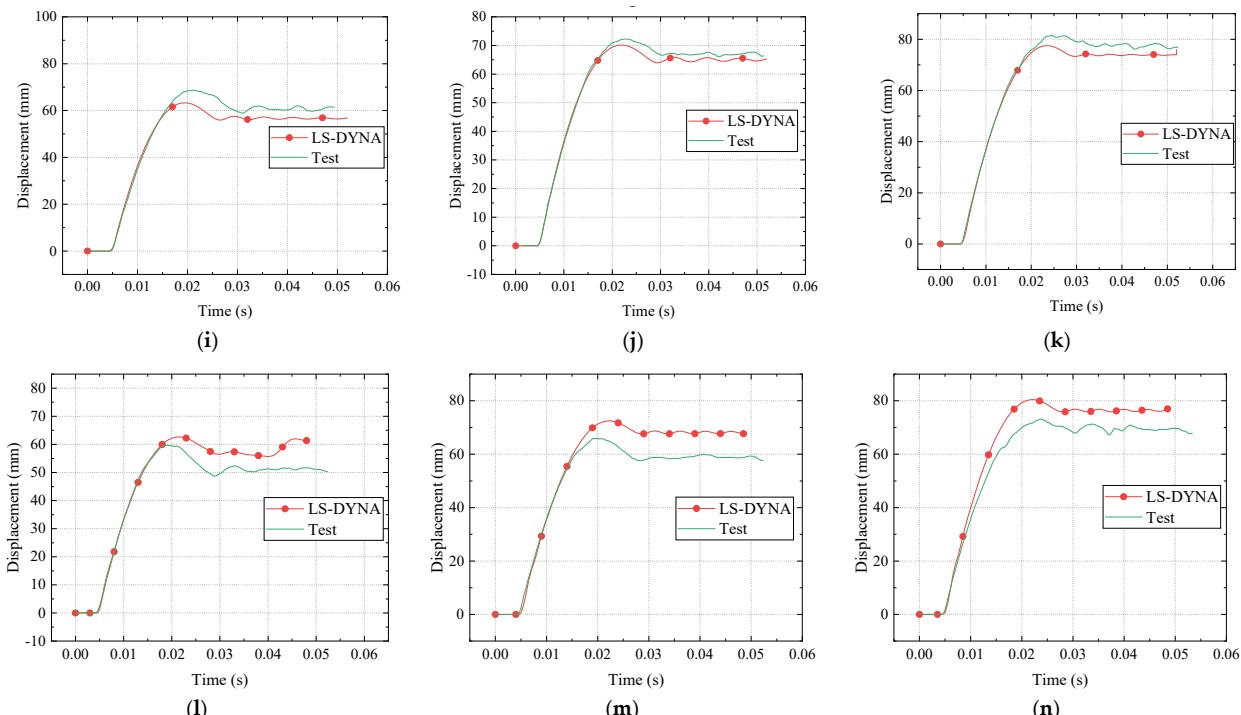

**Figure 6.** Mid-span deflection–time history curve. (**a**) RC; (**b**) RC0.3; (**c**) H-DS; (**d**) H-DS0.3; (**e**) H-DS0.5; (**f**) S-ST; (**g**) S-ST0.3; (**h**) S-ST0.5; (**i**) R-ST; (**j**) R-ST0.3; (**k**) R-ST0.5; (**l**) S-DS; (**m**) S-DS0.3; (**n**) S-DS0.5.

It can be seen from Figures 5 and 6 that in these composite columns, the existence of axial force reduces the impact resistance of specimens within the studied ranges. S-DS columns provide better impact-resistant performances; this is because the existence of the internal and external steel tubes enhances the effect on concrete constraints, thus enhancing the impact resistance of the component.

Figure 7 shows the difference between the numerical simulation and experimental results under various conditions. It can be seen that the difference between most numerical simulation results and experimental results is within 15%. Thus, a good agreement is generally achieved between the predictions and the test results. In Figure 7, $F_{stab}$ is the average of the impact force in the platform stage, $t$ is the time from impact initiation to complete unloading and $\delta$ is the global residual deformation at mid-span.

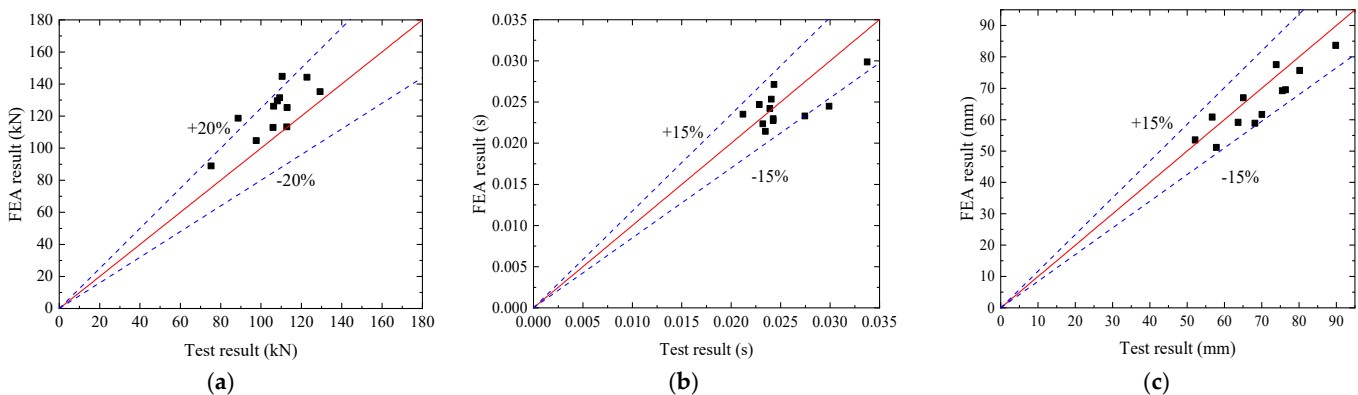

**Figure 7.** Comparisons between the plateau values ($F_{stab}$), load durations ($t$) of the impact forces and the mid-span deflections ($\Delta$). (**a**) Plateau value ($F_{stab}$); (**b**) Load duration ($t$); (**c**) Mid-span deflection ($\Delta$).

## 4. Impact Mechanism Analysis

### 4.1. The Impact Process

Figure 8 shows impact force ($F$)–, mid-span deflection ($\Delta$)–, drop hammer impact velocity ($V_0$)– and component mid-span velocity ($V_m$)–time histories. In order to facilitate the study of the characteristics of the impact process, the curves are normalized to be dimensionless by $F/F_{max}$, $\Delta/\Delta_{max}$, $V_0/V_{0max}$ and $V_i/V_{imax}$, respectively. The drop hammer velocity ($V_0$) and specimen velocity ($V_i$) are negative in the downward direction. Similarly, deflection ($\Delta$) is defined as positive in the downward direction. Although the sections of the specimens were different and the axial forces applied were different, the impact force–time history curves of the specimens had similar stages throughout the impact, which are: peak stage, plateau stage and attenuation stage. Due to the limitation of article length, this paper only presents the normalized time history curves of each specimen without axial force. The stages are summarized as follows:

(1) Peak stage (OA): When the drop hammer touches the specimen, the impact force increases rapidly from zero to the peak value. The specimen demonstrates a downward velocity which rapidly increases to the peak value, which is equivalent to the impact velocity. At the same time, the velocity of the drop hammer decreases. During this stage, a small lateral displacement occurs in the specimen;

(2) Plateau stage (AB): After the impact force reaches ita peak, the specimen velocity decreases to a certain extent, and the velocity of the drop hammer fluctuates slightly. It is the difference in the two velocities that causes the sharp decrease in the impact force. Then, the drop hammer increases the specimen velocity again, and the corresponding impact force also increases after several fluctuations. After several fluctuations, the value remains stable. Subsequently, the drop hammer velocity stays consistent with the mid-span velocity of the specimen, and gradually decreases to zero;

(3) Attenuation stage (BC): At this stage, the drop hammer and the specimen begin to rebound, and the rebound velocity of the drop hammer is greater than that of the specimen, resulting in a decline in the impact force. When the specimen is separated from the drop hammer, the impact force drops to zero. After the specimen reaches the maximum mid-span displacement, the mid-span displacement of the specimen begins to decrease due to the rebound. With the recovery of elastic displacement, the mid-span displacement of the specimen remains stable and recovers to the residual deflection.

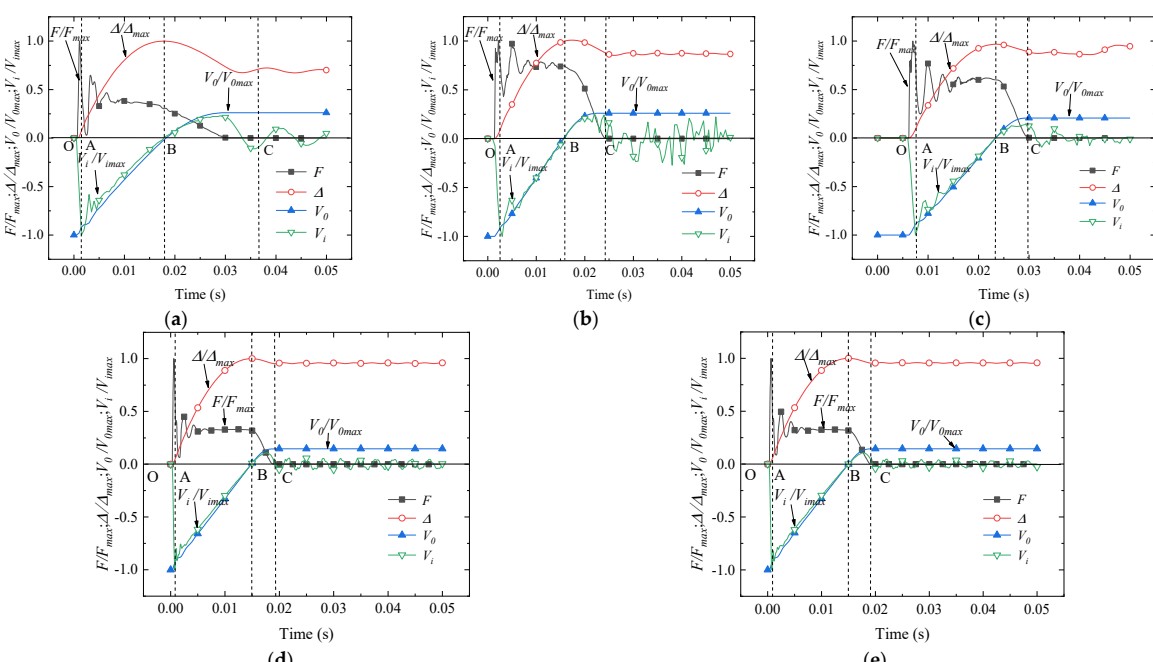

**Figure 8.** Normalized time history curve. (**a**) RC; (**b**) H-DS; (**c**) S-DS; (**d**) S-ST; (**e**) R-ST.

### 4.2. Deformation Mode

Figure 9 shows the equivalent plastic strain of RC, H-DS, S-DS, S-ST and R-ST components impacted by a drop hammer. The red area indicates high plastic strain. It can be seen from the figure that plastic deformation mainly occurs at the impact location and the support locations where there are maximum bending moments. The maximum deformation position of each component is similar, appearing in the middle and lower part of the span and the upper part of the support. The failure modes of internal and external steel tubes and concrete are similar.

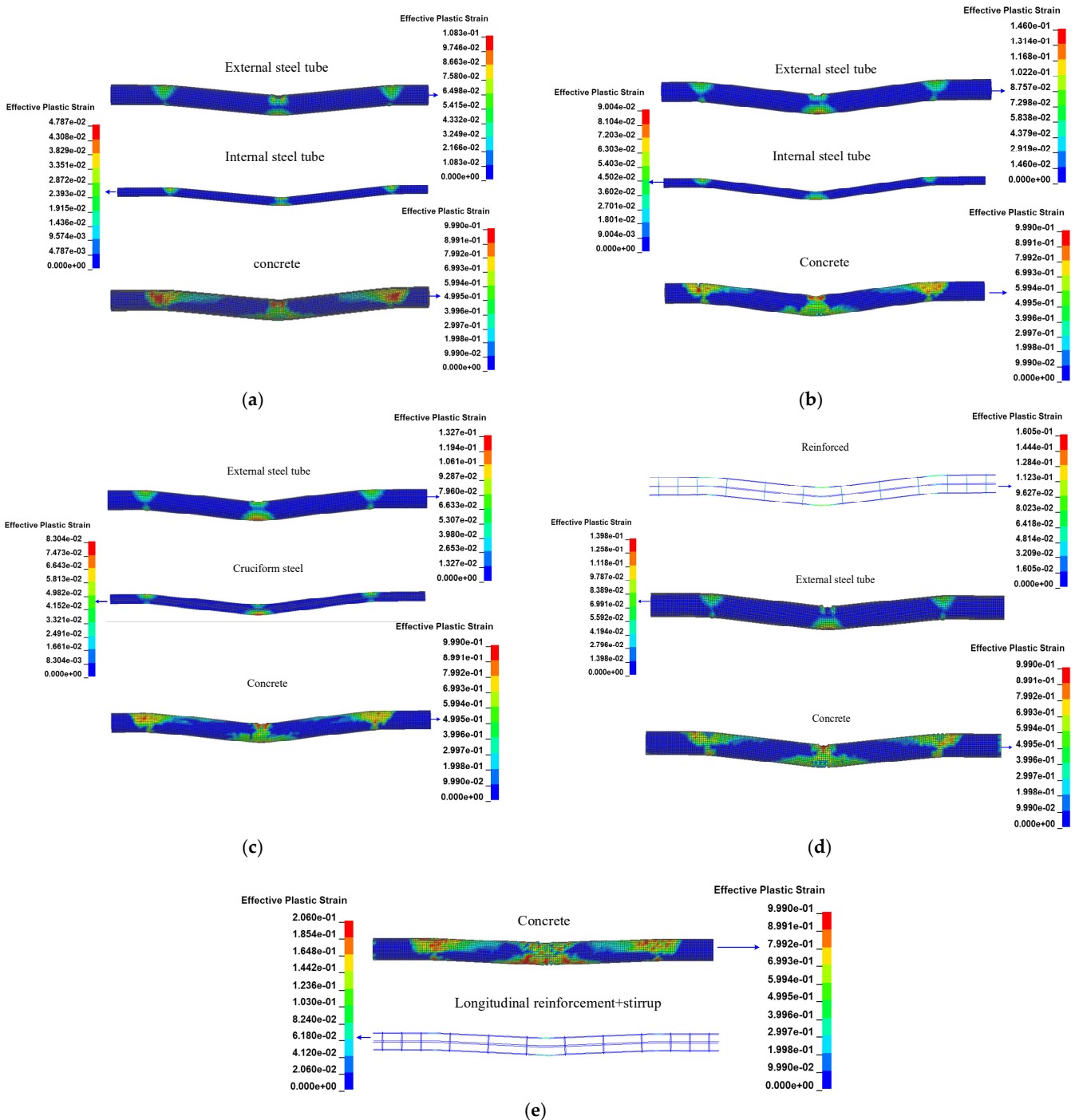

**Figure 9.** Plastic strain of members of different cross sections. (**a**) H-DS; (**b**) S-DS; (**c**) S-ST; (**d**) R-ST; (**e**) RC.

When comparing the different sections, the RC member has the most severe damage with excessive deformation which indicates that the impact resistance of RC columns is limited. For the H-DS member, it was found that the plastic strain of the inner steel tube is much smaller than that of the outer steel tube, which indicates that the energy absorbed by the inner steel tube is smaller than that of the outer steel tube. The maximum plastic strain of the outer steel tube is much smaller than that of the RC column, and the failure zone of concrete is also obviously smaller than that of the RC column, which indicates that the impact resistance of H-DS columns is better than that of RC columns. For S-DS columns, maximum plastic strains of outer and inner steel tubes are increased. For the S-ST member, the maximum plastic strain of the outer steel tube is greater than that of the double H-shaped steel, mainly because the double H-shaped steel is closer to the neutral axis, resulting in a slightly lower maximum plastic strain than that of the outer steel tube. For the R-ST member, the maximum plastic strain of steel bars is greatly improved compared with that of RC columns, which is mainly because the plastic deformation of the external steel tube is the main energy dissipation mechanism, making the deformation of steel bars and concrete smaller than that of RC columns.

*4.3. Energy Dissipation*

Figure 10 shows the internal energy–time history curves of RC, H-DS, S-DS, S-ST and R-ST components. As can be seen from Figure 10, for RC members, the energy absorbed by concrete is slightly greater than that absorbed by steel bars in the early stage of impact, mainly because at this time, concrete produces a large number of cracks due to impact and, thus, absorbs a large amount of energy. In the post-impact period, due to the concrete crack failure resulting in its main energy being absorbed by steel bars, at this time, the energy absorbed by steel bars is far greater than the energy absorbed by concrete. Due to the bending deformation of the specimen, the role of the stirrup is limited and it can hardly absorb energy. For H-DS and S-DS members, their deformation is nearly the same; the energy absorbed is similar. The main energy is absorbed by steel tubes, of which the outer steel tube absorbs most of the energy, about 4.7 kJ (52.24% of the impact energy), followed by concrete, about 2.1 kJ (22.95% of the impact energy), and the inner steel tube absorbs the least energy, about 1.0 kJ (10.93% of the impact energy). For S-ST members, most of the energy is absorbed by the external steel tube, about 5.43 kJ (59.34% of the impact energy), followed by concrete, about 2.23 kJ (24.37% of the impact energy), and the energy absorbed by double H-shaped steel is the lowest, about 1.11 kJ (12.13% of the impact energy). For R-ST members, the main energy dissipation is still absorbed by the external steel tube, about 5.43 kJ (59.34% of the impact energy), followed by concrete, about 2.19 kJ (23.99% of the impact energy), then the energy absorbed by longitudinal reinforcement, about 1.0 kJ (10.93% of the impact energy), and finally the energy absorbed by stirrups, about 2.75 J (0.003% of the impact energy).

Based on the above analysis, the amounts of energy absorbed by steel bars and concrete are roughly equal in the early stage of RC members, and when concrete cracks, steel bars become the main energy dissipation mechanism. For R-ST members and other composite columns, the plastic deformation of external steel tubes is the main energy dissipation mechanism, followed by concrete, and the energy absorption of internal steel tubes, steel bars and double H-beams is the lowest.

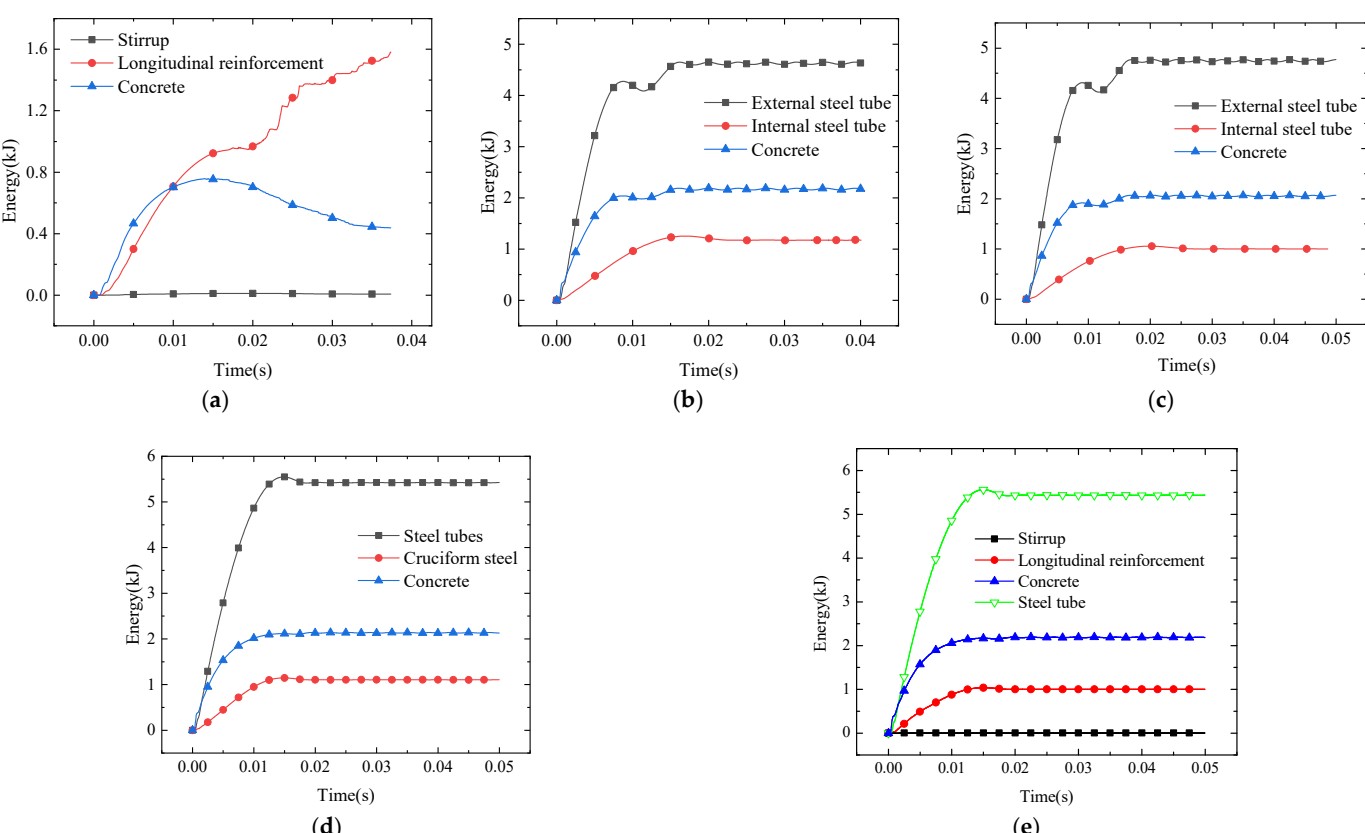

**Figure 10.** Energy–time history curve. (**a**) RC; (**b**) H-DS; (**c**) S-DS; (**d**) S-ST; (**e**) R-ST.

## 5. Parametric Sensitivity Analysis

Following the experimental test, a number of parameters are altered by a percentage to investigate the sensitivity. In the test, the mass is 206.65 kg, the velocity is 9.39 m/s, the compressive strength of concrete is 41.69 MPa, the yield strength of outer steel tubes is 330.1 MPa, the yield strength of inner steel tubes is 330.4 MPa, and the yield strength of steel bars is 529.2 MPa. The yield strength of double H-shaped steel is 295.2 MPa, the diameter–thickness ratio of outer steel tubes is 57, the diameter–thickness ratio of inner steel tubes is 20, the diameter of the longitudinal steel bar is 6 mm and the flange and web width–thickness ratios of double H-shaped steel are 4.67 and 12.25, respectively. These parameters are changed by −40%, −20%, +20% and +40%, and the corresponding changes in impact peak value, impact duration and deflection peak value are compared to conduct sensitivity analysis.

Figure 11 shows the influence of various parameters on the peak impact force, impact duration and deflection peak of H-DS members. The compressive strength of concrete, the diameter–thickness ratios of outer steel tubes and inner steel tubes and the impact velocity have a great influence on the peak impact force. With an increase in the compressive strength of concrete, the peak impact force increases. The diameter–thickness ratio of outer steel tubes and inner steel tubes is negatively correlated with the peak impact force; the greater the diameter-to-thickness ratio, the smaller the peak force. Impact velocity has the greatest influence in the decreasing region, but fluctuates in the increasing region. The impact mass and yield strength of outer and inner steel tubes have little effect on the impact peak force.

The diameter–thickness ratio, impact mass and impact velocity of the outer and inner steel tubes have a certain influence on the impact duration. With an increase in the steel tube diameter–thickness ratio, the wall thickness of steel tubes decreases, and bending stiffness and local stiffness are reduced, resulting in reduced impact force. According to the theorem of momentum, it is known that impact duration increases. The concrete strength

and the yield strength of outer and inner steel tubes have the least influence on the impact duration. With an increase in the yield strength of steel tubes, the impact plateau value of H-DS columns increases, and the impact duration decreases.

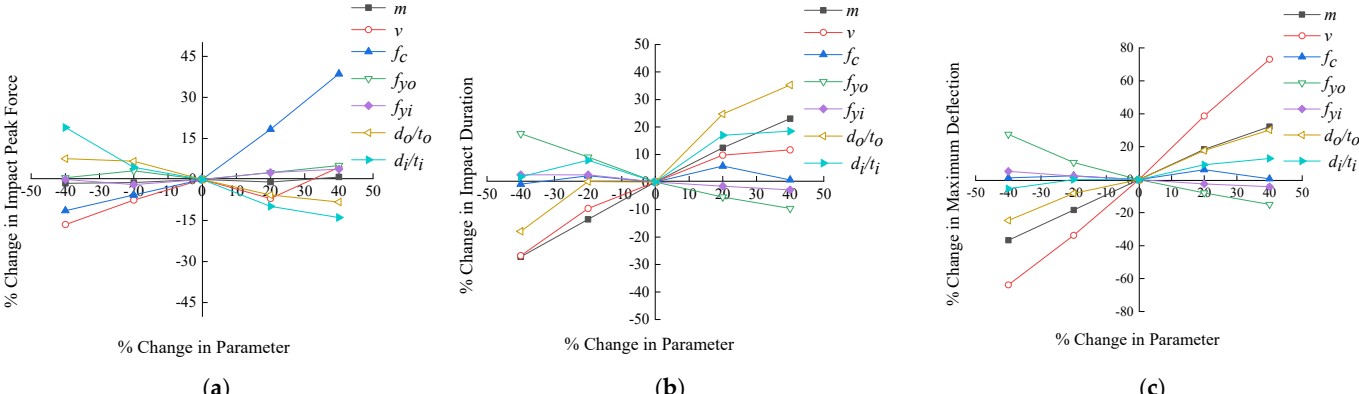

**Figure 11.** Sensitivity of H-DS corresponding to changes in structure-related variables. (**a**) Peak impact force; (**b**) Impact duration; (**c**) Maximum deflection.

Impact velocity has the greatest influence on the peak deflection, followed by impact mass and diameter–thickness ratio of the outer steel tube, diameter–thickness ratio of the inner steel tube, compressive strength of concrete and yield strength of outer and inner steel tubes. This is mainly because the impact energy is proportional to the quadratic power of velocity, so a larger displacement occurs. Changing the diameter–thickness ratio of the outer steel tube also has a great influence on the peak deflection, which is mainly because the bending stiffness of the structural column changes with an increase or decrease in outer steel tube wall thickness, while the inner steel tube has a small influence on the deflection due to the small moment of inertia. Change in other parameters do not effectively affect the bending stiffness of the structure column, so the influence on impact response is small.

From Figure 12, for S-DS columns, the compressive strength and impact velocity of concrete have the greatest influence on the peak impact force; the outer steel tube, the diameter–thickness ratio of the inner steel tube and the compressive strength of concrete have the greatest influence on the impact duration; and the impact velocity, impact mass and the diameter–thickness ratio of the outer steel tube have the greatest influence on the peak deflection.

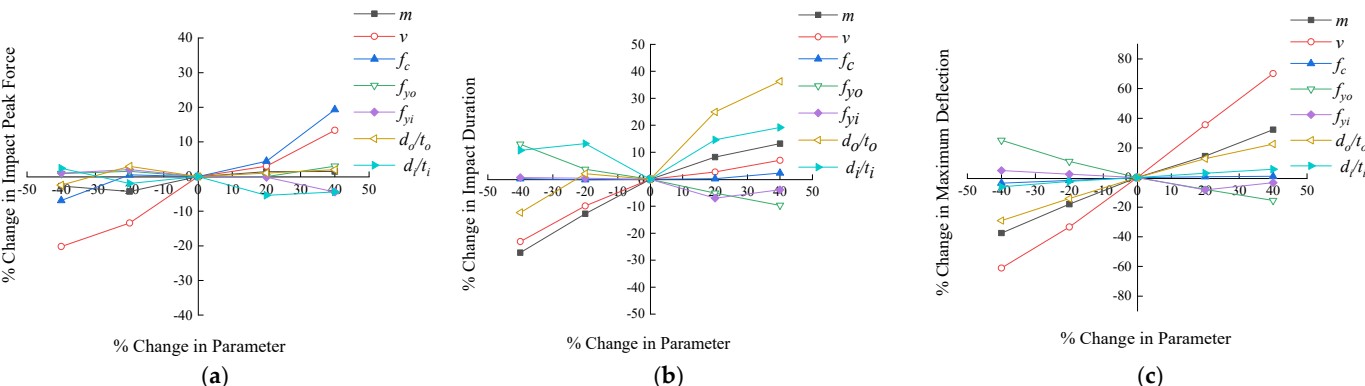

**Figure 12.** Sensitivity of S-DS corresponding to changes in structure-related variables. (**a**) Peak impact force; (**b**) Impact duration; (**c**) Maximum deflection.

Figure 13 shows the influence of various parameters of S-ST columns on impact performance. As can be seen from the figure, the impact velocity and diameter–thickness

ratio of the outer steel tube have greater influences on peak impact value, concrete's compressive strength and double H-shaped steel have slightly greater influences, while steel's yield strength and impact mass have the least influence. This is because the peak impact force is mainly related to the impact velocity and local stiffness. The most effective way to increase the peak impact force is to increase the impact velocity and the stiffness of the external steel tube in direct contact with the drop hammer. Concrete's compressive strength also has a certain influence on the local stiffness. For impact duration, the impact mass and diameter–thickness ratio of the outer steel tube have a major influence, while the yield strength and width–thickness ratio of double H-shaped steel have the least influence; this is because bending stiffness is the most important factor affecting impact endurance. Because the H-shaped section's moment of inertia is small, it is difficult to change the bending stiffness of the section; therefore, it has little effect on impact duration.

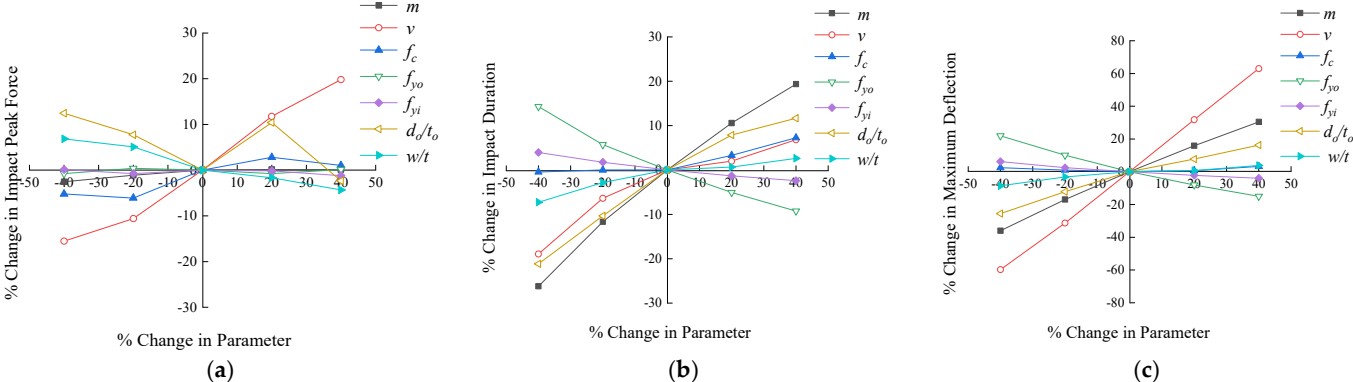

**Figure 13.** Sensitivity of S-ST corresponding to changes in structure-related variables. (**a**) Peak impact force; (**b**) Impact duration; (**c**) Maximum deflection.

Figure 14 shows the changes in energy dissipation of R-ST columns under various parameter changes. As can be seen from the figure, for the peak impact force, the factors that have a greater impact are the diameter–thickness ratio of the outer steel tube and the impact velocity, while other factors have a lesser impact, which is the same as the influence factors of the other three components. For impact duration, the most influential factors are the diameter–thickness ratio of the outer steel tube, impact velocity and impact mass, followed by the diameter of the steel bar, concrete's compressive strength and yield strength of steel bar. For the peak deflection value, the impact velocity and the diameter–thickness ratio of the external steel tube have the greatest influence, followed by the impact mass and the diameter of the steel bar, and the compressive strength of concrete and the yield strength of the steel bar have the least influence.

To summarize, the impact velocity and the diameter–thickness ratio of the outer steel tube have a greater influence on the energy dissipation of the structural column regardless of the cross section, while the influence of the inner steel tube and the double H-beam is the least. The impact velocity mainly increases the impact energy, resulting in greater impact force and impact displacement. The diameter–thickness ratio of the outer steel tube mainly changes the bending stiffness of the section of the structure. Because the outer steel tube is arranged at the outermost part of the section, it provides a larger moment of inertia, thus improving the bending performance of the specimen. Through the above analysis, it can be found that in terms of improving the flexural stiffness of the outer steel tube, changing the diameter–thickness ratio is obviously more effective than changing the yield strength of the component. When the compressive strength of concrete is increased, the interaction with the outer steel tube is strengthened and buckling of the outer steel tube can be prevented. Because the inner steel tube and the double H-shaped steel are closer to the neutral axis, the deformation is smaller than that of the outer steel tube, so the impact deformation cannot be completely deformed, and the impact on the bending stiffness of the section is

less. Therefore, the impact energy dissipation parameters of the specimen are not sensitive to the parameters related to the inner steel tube being near the neutral axis and the double H-beam steel.

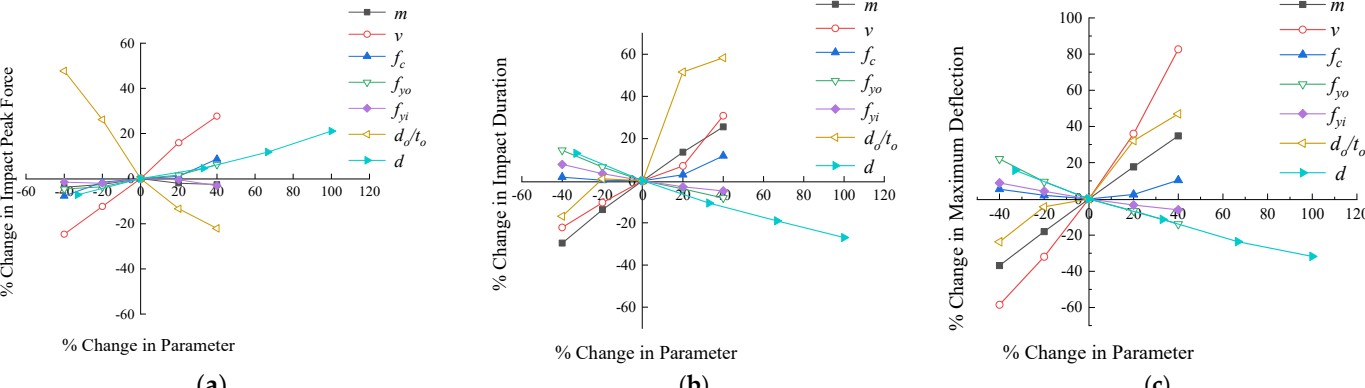

**Figure 14.** Sensitivity of R-ST corresponding changes to change in structure-related variables. (**a**) Peak impact force; (**b**) Impact duration; (**c**) Maximum deflection.

## 6. Conclusions

In this paper, the impact resistance of reinforced concrete columns and composite columns is studied on the basis of numerical simulation. Within the scope of parameter research in this paper, the following conclusions are drawn:

(1) The impact resistance of composite columns is better than that of RC columns. The axial force weakens the impact resistance of the composite column regardless of its cross-sectional form;

(2) In all composite columns, the impact process is similar, and the impact force–time curve shows three stages: peak stage, plateau stage and attenuation stage. Among them, S-DS columns have the best impact resistance, showing a higher impact platform value and a smaller peak deflection under an impact with the same energy;

(3) The simulation results of LS-DYNA fit well with the experimental results. The software is used to analyze the impact process of composite columns, and it is found that the outer steel tube is the main energy dissipation component of these components, which can absorb 50%–60% of impact energy, followed by concrete (about 24% of impact energy), the inner steel tube and the double H-shaped steel (about 12% of impact energy);

(4) LS-DYNA nonlinear finite element software was used to analyze the sensitivity of composite columns. Although the cross-sectional forms of different composite columns were different, the sensitivities of peak impact force, impact duration and peak deflection to parameter changes were similar. The impact velocity and diameter–thickness ratio of the outer steel tube have a great influence on the energy dissipation of the structural column, while the influence of the inner steel tube and the double H-shaped steel is the least, and the influence of concrete is in the middle;

(5) To increase the impact resistance, the best solution is to increase the thickness of the outer steel tube, followed by the yield strength of the outer steel tube and the compressive strength of concrete, and the least economic solution is to improve the impact resistance of components through the mechanical parameters of the inner steel tube.

**Author Contributions:** Conceptualization, X.L. and X.Z.; methodology, T.L.; software, Y.Y.; validation, T.L., X.Z. and R.W.; formal analysis, X.L.; investigation, X.Z.; resources, R.W.; data curation, X.L.; writing—original draft preparation, X.L.; writing—review and editing, X.Z.; visualization, Y.Y.; supervision, X.Z.; project administration, X.Z.; funding acquisition, X.Z. All authors have read and agreed to the published version of the manuscript.

**Funding:** This work was financially supported by the Fundamental Research Program of Shanxi Province (2021030211223022), the Natural Science Young Foundation of Shanxi Province (201901D211178), the National Natural Science Foundation of China (51578274) and the Scientific and Technological Innovation Programs of Higher Education Institutions in Shanxi (2020L0052).

**Institutional Review Board Statement:** Not applicable.

**Informed Consent Statement:** Not applicable.

**Data Availability Statement:** Data is unavailable due to privacy.

**Acknowledgments:** The authors thank Qi Zhang from University of British Columbia for the assistance of proofreading.

**Conflicts of Interest:** The authors declare no conflict of interest.

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
