# Peer review of "Analytical Study on Reinforced Concrete Columns and Composite Columns under Lateral Impact"

_coatings, doi:10.3390/coatings13010152_

Round 1
Reviewer 1 Report
The paper presents a comparison between experiment results for lateral impact on various concrete/steel columns and dynamic simulations using LS-DYNA. The authors show that specific column parameters tend to have a more significant effect on performance during impact that others. The paper is generally well-written but moderate changes are needed to improve readability and clarity.
In lines 40-44, acronyms for the various beam/column types should be added for clarity and to match those used in Figure 1.
Figure 1 is not actually labeled correctly. The caption for Figure 1 refers only to the dimensions of the test specimen. It appears there should actually be a separate figure (Figure 0?) that shows the schematic cross-section for each beam type. In Figure 0, the difference between (a) - R-ST and (e) - RC beams is not clear. Distinguishing features should be described in the text.
The compression ratio is defined on lines 108-109. However, the term is first introduced on line 88. It should be defined when it is first introduced for clarity.
Lines 90-92, when it is said that the amount of steel in each beam is the same is unclear. How can beams with the same outer diameter but with significantly different internal steel reinforcement geometry 'use the same amount of steel'. If the total amount of steel is the same between all beam types, then the dimensions given cannot be correct for all beams. This point needs to be clarified.
In lines 96-97, then reference is made to the dimensions of the cross-shaped steel reinforcement, the reader should be directed to Figure 0f where the geometry is shown.
The current Figure 1 appears to have two versions of the same pair of images (or one is supposed to be Figure 0g and one is supposed to be Figure 1?). One of the duplicates should be removed.
At multiple points in the manuscript, 'Figure X' has been replaced with 'Error! Reference source not found'. These need to be corrected.
On lines 123-129, the authors state that bonding between the concrete and steel is 'good'. This may be true for the experimental samples tested. However, when moving into parametric testing, it is possible that some combination of factors would lead to slip between the steel and concrete during deformation. The authors' model would not capture this transition without allowing for slip in the model. Some acknowledgement of this limitation should be included.
In lines 141-143 reference is made to different calculation methods for concrete compressive strength in the US and China. No citation is provided to justify this parameter or better explain the issue. Additional justification is required.
In Figures 5 and 6, comparison between the different beam geometries would be facilitated by plotting the data with the same axes limits. Granted, this would make it harder to read the graphs for RC beams in Figure 5, for example, which have significantly smaller y values that the other beams. This change is not required but the authors should evaluate whether it would make comparisons between the data easier for the readers.
On line 175, what is the authors' definition of 'better'? At the compression of 0.5, the S-DS geometry provides lower max displacement than any other geometry. However, at a compression ratio of 0.3, the S-DS deformation data is very similar to that of the H-DS geometry. Comparison of the impact forces does not show significant differences between these specimen types. It is unclear what justifies the authors' statement of 'better'.
At line 188, it is not clear if there is a missing figure caption or if the images are intended to be a continuation of Figure 7.
At line 189, the incorrect figure is referenced. The figure is labeled 'Figure 1' but it should be the current Figure 8.
In Figure 8 (line 219). It is unclear what has actually been done in the normalization process. Is the maximum value being used the maximum value for each specimen? If so, shouldn't the curve for each quantity get to 1 at some point on the graph (i.e., doesn't each specimen reach its maximum value of force, velocity, displacement, etc. during the test)? The force and displacement curves make sense from this point of view, but the velocity curves to not.
In the current Figure 10, consistent data labels should be used for Figure 10f. In all other portions of the figure, concrete is labeled with blue triangles. In 10f, this changes to open green triangles. This can be confusing unless the reader is paying close attention.
Data in lines 274-285 should be presented in tabular form along with the associated variables used in Figures 11-14 so that the reader can clearly reference current and modified parameter values.
In Figure 14, why are parameter values of d increasing to 100% included? It is stated that all parameters were varied from -40% to +40%.
Conclusion 4 references effects on the energy dissipation of the structural columns. However, no parametric results are presented for energy dissipation, only max force and max deflection. These do not necessarily scale directly with the energy dissipated depending on the specific shapes of the curves. That is presumably why the authors included Figure 10. Parametric results specifically linked to energy dissipation need to be presented before this conclusion can be made.
In Conclusion 5, it is not clear what is meant by 'impact resistance'. Minimum deflection? Maximum post-impact strength? Something else? It is not clear to the reviewer how the authors intend the data to be interpreted in the context of 'impact resistance'. Additional clarity/definitions are required before this conclusion can be made.
Reviewer 2 Report
Dear Authors,
thank you for the paper focused on the behavior of reinforced concrete and composite columns under lateral impact. In this paper, numerical models in LS Dyna are presented, which are compared with experimental measurements, which were presented in another paper [23]. My comments are:
- Fig. 1 - at the end there is again fig. (a) - if the designation is continued, it should be (h)
- Fig. 1 - the last two pictures (g) and (a) - what is the difference between them? Probably none, so the last one should be deleted,
- line 105 and Tab. 1, there is written mo(kg) = 206.65 kg - for this impact mass, the impact speed is 9.391 m/s, but in tab. 1, another value of 4.43 m/s is given for the RC column - what is valid? Was only the value 9.391 applied? Then you need to correct the value for RC in Tab. 1.
- line 108, the test indicates the force "No", but in tab.1 the force is only "N" - is it the same force? Markings in the text and in the tab should be unified. 1,
- lines 107-108, there is written that were used axial load ratios 0, 0.3, and 0.4 (No/Nu), and from Tab. 1 follows, that there were various No (N), which means that the ultimate forces (resistance) of the samples is not the same, but is different (various), while the greatest bearing capacity is approx. 3 (S-ST) to 3.5 times (S- DS) higher than the lowest (RC column) - is it possible to compare these results and samples? Are incomparable things not being compared?
- line 121 - what does energy28 mean?
- wrong reference in several lines "Error! Reference source not found" (lines 128, 151, 159, 164, 169, 173 (2-times), 179, 221, 246, 247, 286, 315, 321, 335) - links need to be fixed
- line 123 - there is written "good bonding between concrete and steel" - what does "good bonding" mean? Is it perfect bonding without slipping - "rigid contact"?
- Fig. 6, why is the display of RC 0.5 missing?
- Fig. 7 - what does "Plateau" value mean? Also, line 197 - plateau stage - what does "plateau" mean? A plateau is a flat place on the ground that is higher than its surroundings.
- line 188 - these two pictures belong to Fig. 8 on the second page? They must be combined into one Fig. 8. Or are those sieves (d) and (e)? Then they must be deleted.
- line 189 - link to Figure 1 - it should be Figure 8,
- line 219 - there is Fig. 1 - it is wrong, it should be Fig. 8, Fig, 1 is on pages 3-4,
- improve English - "show" or shows" in some cases - "Fig. shows", or "Figs. show",
- Fig. 11, I recommend using the same scale for the x-axis in all three cases - (a) and (b) up to 50%, but (c) is up to 80%, I recommend up to 50% everywhere,
- Fig. 14, - d is the diameter of the steel reinforcement? Wasn't this parameter changed like the other parameters -40%, -20%, +20%, and +40% (given in line 283)? Why was a different change chosen?
- lines 269-371, conclusion (1) - is it possible to compare the impact resistance RC column with different composite columns? These are completely different columns.
Final comments:
1) it is necessary to consider whether this paper belongs to the journal "Coatings". In my opinion, it does not belong to this journal, but to another one, such as "Materials", or "Applied Sciences" and so on. I recommend putting it in another journal.
2) the criterion for the creation of samples of columns was the preservation of the same cross-section, but this represents completely different resistances of the selected cross-sections (tab. 1 shows that the greatest resistance is 3.5-times greater than the smallest) - it is then possible to compare the results with each other and determine which cross-section has the best resistance to impact load? Shouldn't the cross-sections be chosen so that they have the same resistance to vertical loads and then be investigated for impact load? This would mean having different cross-section dimensions, but also different stiffnesses, and the results would be completely different, but very relevant. Or the same effect could be obtained by using different impact loads for individual types of cross sections and using 0, 0.3, and 0.5 from that. Or make a comparison through dimensionless quantities related to the resistance of the given element.
Best regards.

Round 2
Reviewer 2 Report
Dear Authors,
thank you for improving your paper. I still think that it is difficult to compare the selected cross-sections for resistance to impact loads, but by slightly changing the name of the paper it can be accepted. Second thing, it is still necessary to consider whether this paper belongs to the journal "Coatings". In my opinion, it does not belong to this journal, but to another one, such as "Materials", or "Applied Sciences" and so on. I recommend putting it in another journal. But I will leave this decision to the Editor and you. I have no further comments.
Best regards.